# Synergistic Pulmonoprotective Effect of Natural Prolyl Oligopeptidase Inhibitors in In Vitro and In Vivo Models of Acute Respiratory Distress Syndrome

**DOI:** 10.3390/ijms241814235

**Published:** 2023-09-18

**Authors:** Stelios Zerikiotis, Panagiotis Efentakis, Danai Dapola, Anna Agapaki, Georgios Seiradakis, Nikolaos Kostomitsopoulos, Alexios-Leandros Skaltsounis, Ioulia Tseti, Filippos Triposkiadis, Ioanna Andreadou

**Affiliations:** 1Laboratory of Pharmacology, Faculty of Pharmacy, National and Kapodistrian University of Athens, 157 71 Athens, Greece; szerik@pharm.uoa.gr (S.Z.); pefentakis@yahoo.com (P.E.); ddapola@gmail.com (D.D.); seiradakis.g@gmail.com (G.S.); 2Histochemistry Facility, Biomedical Research Foundation of the Academy of Athens, 115 27 Athens, Greece; agapaki@bioacademy.gr; 3Laboratory Animal Facility, Centre of Clinical, Experimental Surgery and Translational Research, Biomedical Research Foundation of the Academy of Athens, 115 27 Athens, Greece; nkostom@bioacademy.gr; 4Section of Pharmacognosy and Natural Product Chemistry Faculty of Pharmacy, National and Kapodistrian University of Athens, 157 71 Athens, Greece; skaltsounis@pharm.uoa.gr; 5Uni-Pharma S.A., 145 64 Athens, Greece; jtsetis@uni-pharma.gr; 6Department of Cardiology, University General Hospital of Larissa, 413 34 Larissa, Greece; ftriposkiadis@gmail.com; 7Faculty of Health Sciences, University of Thessaly, 413 34 Larissa, Greece

**Keywords:** acute respiratory distress syndrome, lipopolysaccharide, natural products, rosmarinic acid, chicoric acid, epigallocatechin-3-gallate, gallic acid, prolyl oligopeptidase

## Abstract

Acute respiratory distress syndrome (ARDS) is a highly morbid inflammatory lung disease with limited pharmacological interventions. The present study aims to evaluate and compare the potential pulmonoprotective effects of natural prolyl oligopeptidase (POP) inhibitors namely rosmarinic acid (RA), chicoric acid (CA), epigallocatechin-3-gallate (EGCG) and gallic acid (GA), against lipopolysaccharide (LPS)-induced ARDS. Cell viability and expression of pro-inflammatory mediators were measured in RAW264.7 cells and in primary murine lung epithelial and bone marrow cells. Nitric oxide (NO) production was also assessed in unstimulated and LPS-stimulated RAW264.7 cells. For subsequent in vivo experiments, the two natural products (NPs) with the most favorable effects, RA and GA, were selected. Protein, cell content and lipid peroxidation levels in bronchoalveolar lavage fluid (BALF), as well as histopathological changes and respiratory parameters were evaluated in LPS-challenged mice. Expression of key mediators involved in ARDS pathophysiology was detected by Western blotting. RA and GA favorably reduced gene expression of pro-inflammatory mediators in vitro, while GA decreased NO production in macrophages. In LPS-challenged mice, RA and GA co-administration improved respiratory parameters, reduced cell and protein content and malondialdehyde (MDA) levels in BALF, decreased vascular cell adhesion molecule-1 (VCAM-1) and the inducible nitric oxide synthase (iNOS) protein expression, activated anti-apoptotic mechanisms and down-regulated POP in the lung. Conclusively, these synergistic pulmonoprotective effects of RA and GA co-administration could render them a promising prophylactic/therapeutic pharmacological intervention against ARDS.

## 1. Introduction

Acute respiratory distress syndrome (ARDS) is a life-threatening pathology characterized as severe acute respiratory failure resulting from acute lung injury (ALI). The main features of ARDS include a non-cardiogenic, pulmonary interstitial and alveolar edema and persistent refractory hypoxemia [1], due to the diffuse inflammation and increased microvascular permeability, caused by the damage of alveolar epithelial cells and capillary endothelial cells of the lungs. Over the years and according to the Berlin definition of ARDS, ARDS is recognized as a significant pulmonary complication with a global impact [2] and associated with considerable rates of morbidity and mortality [3,4] without effective pharmacological therapies [1,2]. Due to the global pandemic of COVID-19, ARDS attracted the interest worldwide in recent years [5].

The development of ARDS is commonly attributed to major causal factors that can be either direct (intrapulmonary), such as in the case of bacterial, viral or fungal pulmonary infections, smoking and inhalation of toxic gases, or indirect (extrapulmonary) like sepsis, severe trauma and acute pancreatitis [6,7]. The acute phase of the disease is evident from the influx of proteinaceous fluid through the disrupted alveolar–capillary barrier, the generation of an inflammatory cascade of cytokines, referred as cytokine storm, leading to the apoptosis and necrosis of the lung epithelium and endothelium [8] and the recruitment of neutrophils and macrophages to the site of infection [9]. These events, along with the formation of eosinophilic hyaline membranes in the alveolar space, constitute the diffuse alveolar damage (DAD), which is regarded as a histological hallmark of ARDS [10] and result in impairment of gas exchange in the alveoli [1,11].

Infiltration of polymorphonuclear leukocytes (PMNs) to interstitial and alveolar spaces also contributes to the pathology. The extensive secretion of proinflammatory cytokines and chemokines, such as tumor necrosis factor alpha (TNF-α) and interleukins such as IL-6, IL-1β, IL-12, cause the recruitment of macrophages/monocytes and neutrophils [12]. The formation of inflammatory molecules, such as reactive oxygen (ROS) and nitrogen (RNS) species, adhesion molecules such as vascular cell adhesion molecule-1 (VCAM-1), intercellular adhesion molecule-1 (ICAM-1) and transcription factors such as nuclear factor kappa B (NF-κB) and activator protein 1 (AP-1), is crucial in ARDS pathogenesis [13]. In this adverse progression of the disease, several pathways are known to be involved, such as the Toll-like receptor 4 (TLR-4)/NF-κB/Mitogen-activated protein kinase (MAPK) pathway (through extracellular signal-regulated protein kinases 1 and 2 (ERK1/2), c-Jun N-terminal kinase (JNK) and p38 kinase), as well as the NLR family pyrin domain containing 3 (NLRP3) and Janus kinase/signal transducers and activators of transcription (JAK/STAT) signaling pathways [14].

Recent studies have also demonstrated the involvement of the cleaving enzyme prolyl oligopeptidase/endopeptidase (POP/PREP) in ALI [15]. The pro-inflammatory nature of POP is attributed to its ability to generate the neutrophil chemoattractant matrikine proline-glycine-proline (PGP) from collagen fragments [16] with the concerted action of metalloproteinases. POP and PGP are both expressed in pulmonary macrophages, epithelial cells, and neutrophils, with the latter being an abundant source of PGP-producing proteases [17,18]. This neutrophilic-driven pathway can generate a self-perpetuating cycle of inflammation, thus linking POP to various pulmonary conditions, such as chronic obstructive pulmonary disease (COPD) and cystic fibrosis [18]. Prior studies have also demonstrated a correlation between POP and COVID-19. High concentrations of POP combined with the downregulation of angiotensin-converting enzyme 2 (ACE2) have been observed in COVID-19; thus, a reduction in the enzyme’s activity could protect against lung injury [17,19].

Previous therapeutic efforts against ARDS such as corticosteroids, nitric oxide (NO) inhalation and dasatinib have been considered ineffective to a greater or lesser extent, because of controversial results regarding the reduction in mortality rate or uncertain risk-to-benefit ratio. As a result, no direct-acting pharmacotherapy has been identified and treatments in clinical practice are still limited mainly to supportive care [1]. For this reason, the discovery of novel molecules that are able to moderate the uncontrolled inflammatory response, without, at the same time, presenting serious side effects to the patients due to other forms of toxicity, is of great translational value and significance. In our study, specific polyphenolic natural compounds were selected due to their anti-inflammatory and anti-oxidant effects and primarily due to their ability to inhibit POP [14,20,21]. Thus, rosmarinic acid (RA), chicoric acid (CA), epigallocatechin-3-gallate (EGCG) and gallic acid (GA) were selected as potential targets against ARDS. The aforementioned natural products (NPs) can be found in many medicinal plant species as well as in fruits and edible herbs, and are known for their anti-bacterial, anti-angiogenic, anti-oxidant and anti-inflammatory effects [22,23,24]. 

In this experimental project, an LPS-induced ARDS mouse model was used as a reliable model of the disease [25]. LPS, or bacterial endotoxin, is a major constituent of Gram-negative bacterial cellular wall that, according to previous studies [26], can generate an ARDS-like acute neutrophilic inflammatory response by initiating cytokine storm, facilitating the vigorous migration of PMNs to the lung and regulating neutrophil chemotaxis [25,27]. The purpose of this study was to examine the prophylactic effect of the selected NPs against LPS-induced ARDS, investigate the molecular mechanisms and pathways involved in their action and evaluate their role as potential molecule candidates against ARDS.

## 2. Results

### 2.1. LPS and NPs Did Not Exert Cytotoxicity in RAW264.7 Cells and MLECs

Initially, the effects of NPs (0–50 μM) on the viability and proliferation of unstimulated and LPS-stimulated macrophages, as well as the effect of LPS on primary mouse lung epithelial cells (MLECs) viability, were investigated. Following a 24 h incubation, LPS administration did not show appreciable cytotoxic activity, as no significant difference in cell viability was observed between the control group and the LPS-treated group on both RAW264.7 and MLECs cells (Figure 1a,b). Administration of NPs on both untreated cells (Figure 1c) and LPS-stimulated cells (Figure 1d) also exerted no significant effect on cell viability.

### 2.2. GA Reduced Nitrite Production in LPS-Stimulated RAW264.7 Cells

Nitrite levels released by RAW264.7 macrophages were measured as an indirect index of NO, following LPS-induced inflammation. Treatment with LPS caused the activation of macrophages and potentiated a significant increase in nitrite production (*p* < 0.0001, Figure 2a). In the absence of LPS, no significant alteration on basal nitrite levels was induced following NPs administration in RAW264.7 cells (Figure 2b). On LPS-stimulated macrophages, administration of GA reduced nitrite production in a dose-dependent manner, with the highest concentration of GA (50 μM) leading to a statistically significant reduction (*p* < 0.05). RA, CA and EGCG, in the presence of LPS, demonstrated no statistically significant inhibitory effect on LPS-activated cells (Figure 2c).

### 2.3. NPs Suppressed the LPS-Elevated Expression of Pro-Inflammatory Mediators in RAW264.7 Macrophages and in Primary Lung Epithelial and Bone Marrow Cells

In order to examine LPS-induced inflammation and the ability of NPs (50 μM) to target and moderate this response, we additionally performed RT-PCR experiments, and the gene expression of key pro-inflammatory mediators which participate in the ARDS-induced cytokine storm was investigated. A significant increase was observed in the mRNA levels of the pro-inflammatory cytokines *Tnf-α*, *Il-6*, *Il-12*, and the *Pop* in RAW264.7 cells, *Il-6*, *Il-12* and chemokine (C-C motif) ligand 2 (*Ccl2*) in MLECs, and *Tnf-α*, *Il-12* and macrophage colony-stimulating factor (*M-csf*) in primary murine bone marrow cells (BMCs) after LPS challenge compared to the levels in the relative control groups (Figure 3a–c).

RA and GA demonstrated the most significant reduction in the mRNA levels of the key pro-inflammatory mediators, namely *Tnf-α*, *Il-6* and *Il-12*, compared to the control group in RAW 264.7 cells (Figure 3a), while at the same time, the two NPs achieved to downregulate the gene levels of *Pop*. In MLECs (Figure 3b), all four NPs caused a decrease in the expression of the levels of *Il-6* and *Ccl2*, with GA exhibiting the greatest reduction. In BMCs (Figure 3c), *Il-12* expression was significantly decreased by CA and EGCG (*p* < 0.01), while RA and GA led to a significant decrease in *Tnf-α* and *M-csf*. From the overall comparative analysis of the four natural products, RA and GA demonstrated the most significant reduction in pro-inflammatory gene expression and seemed to exert most favorable anti-inflammatory properties. Therefore, we selected these two NPs for the conduction of our in vivo experiments.

### 2.4. RA, GA, and Their Co-administration Alleviated LPS-Induced ARDS In Vivo

At first, in order to assess the degree of tissue inflammation, alterations in the influx of inflammatory cells and proteins were measured in the bronchoalveolar lavage fluid (BALF) of our experimental mice 24 h after LPS treatment. LPS exposure led to a significant increase in the BALF cell and protein concentration compared to the data in the control group (*p* < 0.01), while RA and GA, both individually and as a mixture, significantly reduced cell content and protein levels in BALF (*p* < 0.05, Figure 4a,b).

### 2.5. Co-Administration of RA and GA Decreased LPS-Elevated Lipid Peroxidation Levels in BALF

ROS-mediated damage was reflected indirectly by the concentration of MDA in lungs and BALF as a marker of lipid peroxidation [28]. MDA levels were significantly increased in BALF following LPS stimulation when compared to the control group (*p* < 0.05), in contrast to lung tissue, where no significant changes were detected between groups (Figure 4c). Treatment with NPs caused a lowering of MDA levels in BALF, which was statistically significant only after the co-administration of RA and GA (*p* < 0.05) (Figure 4d).

### 2.6. RA, GA, and Their Co-Administration Ameliorated Respiratory Capacity of LPS-Challenged Mice

In addition, the effect of RA, GA, and their co-administration on the highest intrapulmonary pressure during inhalation was measured 24 h post treatment as a marker of LPS-induced lung stress. Upon LPS administration, P_IP_ was significantly increased compared to the control group (*p* < 0.05). Administration of RA and GA caused a significant reduction in P_IP_ levels, with greatest effects observed in the GA and RA + GA groups (*p* < 0.01) (Figure 4e).

### 2.7. RA, GA, and Mainly Their Co-Administration Alleviated LPS-Induced Histologic Changes in Lung Tissues

Lung histologic alterations were examined based on four main parameters of ARDS pathogenesis: hemorrhage, inflammatory cell infiltration, broncheoli injury and vascular intima hypertrophy [29,30]. A score from 0 to 3 (absent, mild, moderate, severe) was assigned to each one. All the aforementioned pathologic alterations were observed to the different groups of our experimental protocol, except the group that received the mixture of the two natural products, where RA and GA seemed to completely abrogate hemorrhage in lung tissues (Figure 4f). According to our results regarding the LPS-treated group, exacerbation of these parameters and damage to bronchioles become evident, with the changes in cumulative histology score being statistically significant (*p* < 0.001) compared to the control group (Figure 4g). Individual treatment of GA significantly ameliorated these changes (*p* < 0.05), while a co-administration of RA and GA brought about the greatest improvement compared to the other groups (*p* < 0.001). 

### 2.8. RA, GA and Their Co-Administration Suppressed the Activation of Key LPS-Induced Inflammation and Apoptotic Mechanisms

Since the expression of key molecules (such as IL-6 and Nf-κΒ) has already been previously studied and it has been demonstrated that protein levels of these inflammatory mediators return to basal levels at 24 h after LPS administration [25], we chose to examine specific pathways linked to inflammation, vascular damage and the apoptosis that are relevant to the selected timepoint and check the potential protective effect of our NPs. Therefore, the expression levels of adhesion molecules (VCAM-1, ICAM-1) and inflammatory mediators (inducible Nitric oxide synthase (iNOS), myeloid differentiation primary response 88 (MyD88)) were investigated via Western blotting. Protein expression of VCAM-1 was significantly increased by LPS (*p* < 0.05), while co-administration of RA and GA caused a significant reduction in VCAM-1 expression (*p* < 0.05) (Figure 5a). Also, a slight reduction in ICAM-1 expression after co-administration of RA and GA in the lung tissues of LPS-treated mice was observed but did not reach statistical significance (Figure 5a). LPS elevated the expression of iNOS that was significantly decreased by RA and GA separately (*p* < 0.01), with their co-administration leading to the greatest reduction (*p* < 0.001). In contrast, neither LPS nor our natural products seemed to provoke any statistical alterations in Myd88 protein expression (Figure 5b).

The potential involvement of apoptotic mechanisms in LPS-induced ARDS was also investigated. Phosphorylation of protein kinase B (Akt) and the expression of anti-apoptotic molecule Bcl-xL (BCL2- Associated X/B-cell lymphoma-extra-large) was elevated following NP treatment. Additionally, treatment with NPs and most favorably their co-administration reversed this LPS-induced deleterious outcome, as the expression of pro-apoptotic molecule Bax, Bax/Bcl-xL ratio and protein levels of cleaved caspase-3 were decreased. Co-administration of the RA and GA appeared to lead to the greatest suppression of pro-apoptotic signaling in the lungs as shown by the significant decrease in Bax/Bcl-xL and a cleaved caspase-3 ratio (*p* < 0.05) (Figure 5c).

### 2.9. RA and GA Co-Administration Reduced LPS-Elevated POP Expression

Finally, as the selection of the NPs aimed at the inhibition of POP, we tested the expression of the enzyme in the LPS-challenged mice in the presence and absence of the NPs. POP levels were significantly increased upon LPS exposure (*p* < 0.05), and only the co-administration of the two NPs resulted in a statistically significant decrease in POP levels (*p* < 0.05) compared to the LPS-treated group (Figure 5d).

## 3. Discussion

Herein, we recruited an in vitro and an in vivo approach in order to successfully recapitulate the complex pathomechanism of ARDS and for first time propose a translational insight in the potential pulmonoprotective properties of natural POP inhibitors. According to our in vitro results, RA and GA seemed to exert the greatest protective effects against LPS-induced inflammation. In LPS-stimulated RAW 264.7 macrophages, GA treatment significantly decreased nitrite levels in LPS-challenged cells, thus exerting anti-inflammatory activity, which is in consistence with prior studies [31]. Nitrite is the stable oxidation product of NO and is indicative of NO levels produced by cells [32]. The observed inhibition of NO production could be beneficial to the affected lung tissue, as high levels of NO have been previously associated with acute or chronic inflammation [33]. Previous findings have suggested that LPS-stimulated NO release from RAW264.7 macrophages is associated to NF-κB -dependent regulation of iNOS expression [34]. Although nitric oxide (NO) is an important part of the host defense mechanism and plays physiological roles in several cellular processes, the formation of RNS after excessive NO production is considered the main NO contribution in both acute and chronic inflammation [35]. NO might have distinct roles in different stages of pulmonary inflammatory diseases, as it has been reported to be deleterious and pro-inflammatory in acute and severe stages of the disease but also acting as a vasodilator with potentially protective and anti-inflammatory properties in more stable conditions. These contradictory effects of NO may be due to the localization, time and the amount of NO synthesis, the acute or chronic character of the immune response or the type of lung injury model and the absence of specificity for the different NOS isoforms inhibitors used within the studies [36]. Moreover, despite the alleged harmful effects of NO that have been demonstrated, NO inhalation has been suggested as a potential therapeutic treatment for ARDS patients in order to reduce neutrophilic inflammation and improve overall gas exchange by acting directly to the pulmonary vasculature located in the vicinity of the ventilated lung. However, results of NO efficacy at a clinical level are controversial, as it is associated with limited to no improvement in oxygenation and mortality rates of ARDS patients [37]. Therefore, additional pre- and clinical studies are required in order to decipher the complete pulmonoprotective or deleterious potential of NO in lung diseases. 

Besides macrophages, we also examined the effect of NPs on primary cell lines isolated from C57BL/6J mice that play an important role in ARDS pathogenesis, such as MLECs and BMCs. Neutrophils constitute the most abundant cellular population in the bone marrow where their G-CSF (granulocyte colony-stimulating factor)-regulated production takes place [38,39]. The mature neutrophils that remain as a reserve in the bone marrow and released upon the inflammatory stimulus, are the first cells recruited from circulation to the site of infection [40]. As neutrophils are an important source of PGP-producing proteases, like POP [17], we considered that inhibition of the enzyme by our NPs could be an interesting approach in the management of inflammation and the provoked lung injury.

Inflammation is considered to play a pivotal role in ARDS and could be directly or indirectly responsible for alveolar epithelium and microvascular endothelium damage in lungs, which are the primary events in its pathogenesis [41]. Relative macrophage-regulated levels of pro- (e.g IL-6, IL-1β, TNF-a) and anti-inflammatory cytokines (e.g., IL-4, IL-10) have been demonstrated to correlate with the severity and prognosis of lung injury [42]. Oxidative stress and apoptosis are also implicated in the progression of ARDS [43]. Regarding pro-inflammatory mediators, RA and GA significantly reduced their gene expression in vitro. Particularly in RAW264.7 macrophages, RA and GA caused a significant decrease in the LPS-elevated mRNA expression of *Il-6, Il-12, Tnf-α* and *Pop*. Previous in vitro and in vivo studies have also demonstrated the downregulation in the expression of key pro-inflammatory cytokines and chemokines, that strongly affect the acute inflammatory response from RA and GA, mainly through the suppression of the TLR4-mediated NF-κB signaling pathway [20,44,45,46]. IL-6 is believed to play a pleiotropic role in inflammation and the cytokine storm; thus, an early inhibition of its expression could ameliorate the severity of inflammation [47]. In this context, since IL-6 plays an important role in cytokine release syndrome, one of the treatment strategies tested against moderate to severe COVID-19 disease was tocilizumab, which is a recombinant humanized monoclonal antibody acting as an antagonist on the IL-6 receptor [48]. Despite some conflicting results and limitations in its usage, several studies have reported beneficial outcomes regarding tocilizumab’s efficacy in many COVID-19 patients [48,49]. Therefore, our natural products that have been shown to possess antioxidant and anti-inflammatory properties and to exhibit a pleiotropic mechanism of action [14,20,22,23,24], without causing immunosuppression or other adverse effects like tocilizumab [48], could serve as a possible translational alternative.

In addition, reduction in *Tnf-α* mRNA levels further reinforces the anti-inflammatory action of the two NPs, as low levels of TNF-α could prevent vascular permeabilization and oedema exacerbation [50]. Additionally, this could also explain the anti-oxidant effects of the NPs, as TNF-α is involved in the reduction in antioxidant production and free radical scavenging [51]. Expression of *Il-12* was also suppressed, which could indicate that events such as the production of Th-1 effector cells that mediate lung injury become hindered [52]. Furthermore, expression of *Ccl2* (monocyte chemoattractant protein-1, MCP-1) was reduced in MLECs while that of M-*Csf* was reduced in BMCs. Given that M-CSF regulates macrophage survival and that both mediators are responsible for macrophage proliferation and recruitment [53,54], the downregulation of their inflammation-elevated expression could normalize the acute inflammatory response.

Subsequently, RA and GA were the two NPs that were selected for in vivo experiments due to their ability to mitigate LPS-induced inflammation in vitro. Intratracheal LPS administration was carried out to initiate an acute influx of inflammatory cells to the lungs [55]. Even though most studies have evaluated RA as a prophylactic treatment, with the compound being administered simultaneously with LPS [56], in our study, for the first time, the NPs were administered 6 h after LPS administration in vivo. This timepoint reflects a harsh timepoint in the progression of ARDS, as at that timepoint, an establishment of uncontrolled inflammation is already observed. Therefore, the results of our in vivo study present, for the first time, the therapeutic and not the prophylactic potential of the natural POP inhibitors RA and GA. The latter increases the translational value of our findings, considering that patients with ARDS or ALI already have an established lung inflammation at the time of the admission [25]. 

During disease progression, the molecular mechanisms implicated in ARDS are differentially activated over time. According to the literature, the main proinflammatory mediators and cytokines such as IL-6, TNF-a, and IL-1β seem to rapidly increase, reaching maximal levels at the first hours after intranasal or intratracheal LPS challenge, followed by a decrease, reaching basal levels, at 24 h [57]. Based on previous findings supporting the notion that cytokine concentration (e.g., IL-6) is at the maximum levels and excessive inflammatory response occurs 6 h post LPS treatment, we selected this timepoint as a harsh time point for the administration of our NPs [25,58]. The experimental duration of 24 h was selected, as neutrophil infiltration into the lungs seemed to reach maximal influx at that time point [25].

RA and GA co-administration seemed to exert the greatest beneficial effects in LPS-challenged mice. LPS-aggravated lung resistance (P_IP_) and cell and protein concentrations in BALF were significantly decreased by the RA and GA mixture. Such results could indicate reduced PMN infiltration, improved lung compliance, and overall minimization of respiratory distress [59]. Regarding oxidative stress, the decrease in LPS-amplified MDA levels in BALF suggests potential anti-oxidant action of NPs through the suppression of lipid peroxidation and neutralization of LPS-generated free radicals [43]. Morphologically, histologic alterations and hallmark symptoms such as lung trauma and oedema were significantly ameliorated, thus confirming pulmonoprotective activity of RA and GA co-administration. An overall protective activity of RA and GA separately against the LPS-induced lung injury has also been supported by prior literature [20,43,53,60], but the therapeutic potential of the two NPs and their co-administration was demonstrated in our research for the first time in an in vivo ARDS model. In addition, an alleviation in hemorrhage in lung tissues because of RA or GA administration has been reported in different models of lung injury [61,62], but in our work, the combination of RA and GA treatment seemed to totally abrogate hemorrhage in the lung tissue of the LPS-challenged mice after histopathologic examination. 

Increased expression of iNOS has been associated in previous studies with both disease severity and progression in acute respiratory distress syndrome [63,64], asthma [65] and chronic obstructive pulmonary disease [66,67]. Upregulation of the expression of cell adhesion molecules, like intercellular adhesion molecule (ICAM)-1 and vascular cell adhesion molecule (VCAM)-1, has been linked to trafficking, adhesion and transmigration of leukocytes in various animal models of inflammatory processes [68]. Our Western blot analysis showed that LPS-elevated expression of the inflammatory marker iNOS and the marker of endothelial activation VCAM-1 reduced following RA and GA administration, indicating that PMN recruitment to the alveolar space might be inhibited [69]. Therefore, the observed decrease in VCAM-1 and iNOS expression in lung tissues indicates that RA and GA administration could exert pulmonoprotective activity at least partially through the modulation of inflammation, leukocyte adhesion and transendothelial migration. In contrast, MyD88 expression remained invariable between groups, which could be in accordance with the results of earlier studies supporting the notion that pulmonary levels of *Myd88* mRNA expression were rapidly elevated to maximal levels at 3 h, while *Myd88* levels were restored to basal levels 24 h after intranasal LPS challenge [57].

Regarding LPS-induced apoptosis, NP-mediated reduction in the Bax/Bcl-xL ratio indicated the decreased concentration of Bax in lung tissues. Members of the Bcl-2 family control the balance for the proapoptotic or antiapoptotic outcome [70] and Bax, a major pro-apoptotic protein of the Bcl-2 family, induces cell death via the intrinsic apoptotic pathway [71]. Initially, it oligomerizes on the outer mitochondrial membrane, increasing its permeabilization [72]. Subsequently, it induces the opening of mitochondrial permeability transition pores (MPTPs) and the subsequent release and translocation of cytochrome c and other macromolecules from the intermembrane space to the alveolar cytosol [71,73]. Apoptosis is thereby initiated following the binding of cytochrome c to the apoptotic protease activating factor-1 (APAF-1), which leads to the activation of caspase-9 and caspase-3 [74]. In this study, the change in the Bax/Bcl-xL ratio to favor Bcl-xL and the decrease in cleaved caspase-3 expression after RA and GA treatment, could therefore suggest that apoptosis was inhibited, as co-administration of the two natural products appeared to prevent the activation of the mitochondrial apoptotic pathway. This preventive effect on the apoptotic procedure mediated by the two natural products and their combination therapy with the inactivation of the mitochondrial intrinsic apoptotic pathway has not been demonstrated in the past in an ARDS model. Apoptosis of pneumocytes (mainly type I epithelial cells) contributes significantly to the progression and manifestation of the acute respiratory distress syndrome [75]. In the next step of the disease, when persistent inflammation and injury occur, a fibroproliferative phase initiates for resolution of the damage followed by an irreversible fibrotic end stage in many patients if counter-regulatory mechanisms cease or fail [75,76]. Therefore, the suppression of LPS-induced apoptosis by RA and GA co-administration at this stage, as we showed in our work, could be a novel and translationally significant finding in the battle against ARDS morbidity.

Expression of survival protein Akt was also increased following RA and GA co-administration. The Akt signaling pathway probably acts as a compensatory mechanism since Akt has been shown to control cell survival, proliferation, and growth through its kinase activity and to be involved in the suppression of proinflammatory and apoptotic events [77]. Akt is a serine–threonine kinase downstream of phosphoinositide 3-kinase (PI3K) which, once phosphorylated, could regulate target proteins downstream, such as the suppression of NF-κB expression. NF-κB activation could lead to the translocation of its p65 subunit towards the nuclear area and to an increased expression of proinflammatory genes [78,79]. Previous studies have also demonstrated the importance of the PI3K/Akt pathway in lung inflammation and ALI/ARDS progression, mainly by regulating the survival of cells during oxidative damage [80]. Activation of the PI3K/Akt pathway was also shown to promote the activation of the BCL2 associated agonist of cell death (Bad) protein, which releases the anti-apoptotic molecule Bcl-xL [81]. As a result, administration of RA and GA could potentially block the activation of the mitochondria-dependent apoptotic pathway and ameliorate lung injury [71].

Additionally, POP gene expression was suppressed by RA and GA in RAW 264.7 macrophages, while their co-administration also decreased the enzyme’s LPS-elevated levels in vivo. Prolyl oligopeptidase (PREP) is a member of the serine protease family with the capacity of hydrolyzing peptides with less than 30 amino acids [82]. The POP-associated production of the collagen-derived neutrophil chemoattractant PGP and the self-sustaining neutrophil-driven constant inflammation linked the enzyme to various inflammatory pulmonary conditions, such as COPD and cystic fibrosis [18]. Previous studies have demonstrated the involvement of POP in inflammation, angiogenesis, oxidative stress and apoptosis, so POP inhibition could be a potential target in order to counteract lung injury [15]. A number of POP inhibitors, such as the selective inhibitor of POP, KYP-2047 (4-phenyl-butanoyl-l-prolyl-2(S)-cyanopyrrolidine) and valproic acid, have been developed and extensively studied in various in vitro and in vivo models of inflammatory diseases with beneficial effects against these mechanisms that the POP enzyme is involved [15,83,84]. In addition, a clinical trial of the selective phosphodiesterase-4 inhibitor, roflumilast, demonstrated that the alleviation in pulmonary inflammation after drug administration in patients with moderate-to-severe COPD was attributed to a decrease in pulmonary POP, Acetyl-PGP (AcPGP) levels, and neutrophilic inflammation [85]. Both our natural products have been previously recognized as POP inhibitors, and thus, the blockage of POP activity could have beneficial outcomes in regulating inflammation [20,86] and be partially responsible for lung protection. To our knowledge, this is the first time that a decrease in POP expression from both NPs has been demonstrated at this timepoint of administration in an in vivo ARDS model.

Taking our results together, it seems that the combination of GA and RA has the most remarkable effects against the LPS-induced lung injury, while the GA or RA individually do not lead to significant improvements in the phenotype, suggesting a synergistic effect of the two polyphenols. We can speculate that the different substances may affect the inflammatory process at a different stage or in distinct cell populations related to the LPS induced injury. GA had a significant impact on macrophages, according to our in vitro observations, since it reduced inflammation markers and the gene expression of stimulatory and chemoattractant factors such as *Mcsf* and Monocyte chemoattractant protein-1 (*Mcp-1/Ccl-2*). The activation of macrophages is related to the initial phases of lung inflammation and injury [87], and therefore we assume that GA may act early in ARDS pathogenesis. Although RA does not prevent polarization of macrophages as indicated by NO release, we point out the significant reduction in *Il12* gene expression in Raw264.7 and lung epithelial cells. This decrease suggests that RA may play a significant role later in the inflammatory cascade during ARDS progression by a regulation of T-cell differentiation and coordination of innate and adaptive immunity [88]. Therefore, the combination of GA and RA could act synergistically to combat the inflammation in different ways. Finally, GA alone, but not RA, leads to a significant reduction in apoptosis in the lung tissue, which is also more remarkable in their combination and could be attributed to the antioxidant properties of their combination. Moreover, GA could potentially have a more decisive contribution to the protection of pulmonary epithelial cells against apoptosis. Our investigation sheds light on the anti-inflammatory and anti-apoptotic effects of both RA and GA, but due to their pleiotropic effects [14,22], we cannot rule out that other mechanisms of lung protection may be involved. 

A limitation of our study is that we could not correlate the in vivo doses and the in vitro tested concentrations of each natural product, whereas dose and concentration regimens were based on previous reports [20,89]. Moreover, our study lacks clinical proof of the pulmonoprotective effect of the compounds reported or their combination. The better assessment of the pharmacokinetics and pharmacodynamics of our compounds could improve our knowledge concerning the degree of contribution of the two NPs in vivo and probably also shed some light on their synergistic activity. Moreover, since the existing murine lung injury models do not seem to precisely recapitulate all aspects of the human ARDS, the challenge remains to translate these findings and prove the efficacy of our compounds in clinical trials. Even though translatability of animal studies has been unsuccessful in ARDS-related clinical trials until now [73,90], the better identification of the biological pathways involved, the development of novel approaches in NPs administration for improved bioavailability and the selection of an appropriate patient population based on evidence-driven criteria, such as biomarkers, risk factors, phenotypes or concurrent medications of the patients, could be helpful in the achievement of this objective. 

In conclusion, co-administration of the natural inhibitors of the POP enzyme, RA and GA, exerted significant pulmonoprotective effects by suppressing LPS-induced inflammation and apoptotic pathways. The identification of the extent and the exact point prolyl oligopeptidase is implicated in these complicated mechanisms that result in ARDS pathogenesis, could be an interesting aspect for the discovery of novel therapeutic approaches in managing ARDS. Taking into consideration that clinical arsenal lacks specific prophylactic and therapeutic therapies against ALI/ARDS, we herein propose a novel combination of NPs, RA and GA, as a potent therapeutic intervention after the establishment of the disease. The use of NPs in therapy based on their cytotoxicity profile might also contribute to the mitigation of side effects, such as severe gastrointestinal irritations, allergic reactions or immunosuppression (such as steroids and immunotherapy [48,91,92]), while it could be beneficial against long-term morbidity after critical illness. In this study, for the first time, we proposed a new dose and administration regimen for RA and GA, namely that of intraperitoneal injection, which can be employed in future clinical studies investigating their pulmonoprotective potential. Nevertheless, further pre- and clinical studies are required for the establishment of the combination therapy as a potent therapeutic intervention against ALI/ARDS.

## 4. Materials and Methods

### 4.1. Reagents

All four natural products (NPs), rosmarinic acid (RA), chicoric acid (CA), epigallocatechin-3-gallate (EGCG) and gallic acid (GA) and LPS procured from *E. coli* were provided as powder and were dissolved into a liquid form with a Phosphate Buffer Saline (PBS) solution. All reagents were purchased from Sigma Aldrich (Merck KGaA, Darmstadt, Germany via Life Science Chemilab SA and Tech-Line chemicals SA, Athens, Greece), while all antibodies were purchased from Cell Signaling Technology (St. Louis, MO, USA, via Bioline Scientific, Athens, Greece), unless otherwise stated. Compounds were kindly provided from the chemical library of the laboratory of valorization of bioactive natural products, Faculty of Pharmacy, National and Kapodistrian University of Athens, Greece (Table 1).

### 4.2. Isolation of Primary Mouse Lung Epithelial Cells (MLECs)

The procedure of isolation and culture of primary mouse lung epithelial cells (MLECs) was conducted according to the literature [94]. Mouse hearts were initially perfused with PBS while the lungs were intratracheally inflated with a 0.5 mL combined solution of PBS and ethylenediaminetetraacetic / ethylene glycol-bis(β-aminoethyl ether)-N,N,N′,N′-tetraacetic acid (EDTA/EGTA). Lungs were then perfused with 1.73 mg/mL of elastase, excised and stored in Falcon tubes containing elastase. Subsequently, they were incubated at 37 °C for 20 min and their digestion was terminated with Fetal Bovine Serum (FBS) and a 25 μL DNase. The lung parenchyma was separated from the large airways and filtered through a 40× μm strainer. Epithelial cells were then washed with a PBS solution and centrifuged at 300×g for 10 min. The cellular precipitate was stored in 3 mL of starvation Dulbecco′s Modified Eagle′s Medium (DMEM) [94]. MLECs were then seeded in 24-well plates.

### 4.3. Isolation of Primary Mouse Bone Marrow Cells (BMCs)

Isolation of mouse bone marrow cells (BMCs) was performed as previously described [95]. Murine femurs and tibias were isolated and placed onto Petri dishes containing a Roswell Park Memorial Institute (RPMI) 1640 medium. Muscle and residue fibrous tissue surrounding the bones was removed with forceps and the knee joints were cut off. An insulin syringe filled with a 1 mL PBS solution was used to flush the bone marrow content out of each hindlimb until bones turned white in color [95]. Cells were then transferred into a 3 mL PBS solution and centrifuged at 350× *g* for 6 min. The high erythrocyte content of BMCs was eliminated by the resuspension of the pellet in a 2 mL Red Blood Cell (RBC) lysis buffer (NH_4_Cl, 155 mM; KHCO_3_, 10 mM; EDTA, 0.1 mM) for 3 min. Lysis was neutralized with 4 mL of the PBS solution and cells were centrifuged for 6 min at 350 × *g* [95]. BMCs were then stored and seeded in a manner similar to that of MLECs.

### 4.4. Cell Cultures

RAW264.7 cells purchased from Sigma Aldrich (Life Science Chemilab SA, Athens, Greece) were maintained in high-glucose DMEM, supplemented with a 10% heat-inactivated FBS, 1% antibiotics (penicillin/streptomycin) and 5 mM L-glutamine. Cells were re-cultured every 2–3 days when they reached about 75% of confluence. The primary cell lines (MLECs and BMCs) were maintained in an RPMI medium. Cells were seeded in 96-, 24- or 12-well plates depending on the experimental procedure and incubated in a humidified chamber (CB170, BINDER) at 37 °C and 5% CO_2_ to facilitate cell attachment and proliferation.

Cells were either untreated (control group) or treated with either LPS (10 μg/mL) (LPS-treated group), each of the four NPs (RA-treated group, CA-treated group, EGCG-treated group and GA-treated group) or a combination of LPS (10 μg/mL) and each NP (LPS + RA-treated group, LPS + CA-treated group, LPS+EGCG-treated group and LPS + GA-treated group) in different concentrations (5 nM–50 μM).

### 4.5. Evaluation of Cell Viability

After treatment of cells with LPS and/or NPs, cell viability was determined by the mitochondria-dependent reduction in 3-(4,5-dimethylthiazol-2-yl)-2,5-diphenyltetrazolium bromide (MTT) to formazan as previously described [96]. RAW264.7 macrophages and MLECs were incubated with 150 μL of an MTT reagent (12.5 mg/mL) dissolved in DMEM for 3 h at 37 °C in the dark. The purple formazan precipitate was then dissolved in dimethyl sulfoxide (DMSO). Absorbance was measured at 570 nm (background absorbance measured at 680 nm) using a TECAN Infinite M200 PRO plate reader (Tecan Group Ltd., Männedorf, Switzerland).

### 4.6. Assessment of NO Production of RAW264.7 Macrophages

The production of the stable metabolite nitrite was measured in RAW 264.7 culture supernatants via the Griess reaction as an index of the short half-life molecule NO [97]. Our samples (50 μL of cellular supernatants) were incubated with 50 μL of 1% sulfanilamide in 5% H3PO4 for 5 min and then with 50 μL of 0.1% N-(1-naphthyl)-ethylenediamine dihydrochloride (NED) in H_2_O for another 5 min. Absorbance was measured at 550 nm with the TECAN Infinite M200 PRO plate reader (Tecan Group Ltd., Männedorf, Switzerland). Nitrite concentration was back-calculated based on the potassium nitrite (KNO_2_) calibration curve (0–100 μM) which was used as a standard curve for NO_2_.

### 4.7. RNA Extraction, cDNA Synthesis and RT-PCR

After treatment, RAW264.7 macrophages and the primary cell lines were treated with the TRIzol^®^ reagent (Thermo Fischer Scientific Inc., Waltham, MA, USA), and total RNA was extracted via the phenol/chloroform extraction method as described previously [98]. RNA (ng/μL) was measured via the NanoDrop UV–Vis spectrophotometer (Thermo Fischer Scientific Inc., USA). Isolated RNA was reverse transcribed to form cDNA using the FastGene^®^ Scriptase II cDNA kit (Nippon Genetics, Düren, Germany). Real-time PCR was performed with the CFX96 Real-Time PCR Detection System (Bio-Rad, Munich, Germany) using the EvaGreen^®^ fluorescent dye method for qPCR (Biotium Inc., Fremont, CA, USA) according to the manufacturer’s instructions. The primer pairs were designed (Primer-Blast, NCBI, NIH) and synthesized by Eurofins Genomics AT, GmbH, and primer sequences used for the Real-time PCR analysis are presented in Table 2. These primers were used to measure the mRNA expression of the inflammatory mediators *Il-6, Il-12, Tnf-α, M-csf, Ccl2* and enzyme *Pop*. Gene expression of the inflammatory mediators was quantified using the ΔΔC_T_ method. mRNA levels were normalized to those of the housekeeping gene (β-actin), and the target gene mRNA expression of each sample was expressed relative to that of the control.

### 4.8. Animals

A total of 30 (13–14 weeks of age) C57BL/6J male mice (~25 g) were bred and housed at 25 °C in microisolators in the Animal Facility of Biomedical Research Foundation Academy of Athens and were provided food and water ad libitum. Experiments were carried out in accordance with the “Guide for the care and use of Laboratory animals” and with the Directive 2010/63/EU, and the experiments were approved by the Ethics Committee, Veterinary Service of the Prefecture of Athens (Approval No: 124357/15-02-2022) according to ARRIVE guidelines [99].

### 4.9. In Vivo Experimental Protocol of LPS-Induced ARDS Model

For our in vivo experiments, the pulmonoprotective effects of the two NPs that exerted the most favorable properties in vitro, namely RA and GA, were investigated. Lung injury was induced following intratracheal LPS instillation, which is the classic model mimicking the mechanisms of ARDS pathogenesis [25].

C57BL/6 male mice were divided into the following 5 groups of 6 animals each: (a) Control (received 0.9% phosphate buffer saline (PBS) intratracheally (IT)), (b) LPS-treated (2 mg/kg IT), (c) RA-treated (LPS 2 mg/kg IT and RA 20mg/kg intraperitoneally (IP)), (d) GA-treated (LPS 2 mg/kg IT and GA 20 mg/kg IP) and (e) RA + GA-treated (LPS 2 mg/kg IT and RA + GA 20mg/kg + 20mg/kg IP). The dosages of the NPs given to the laboratory animals were selected according to the literature [100,101,102]. All NPs administrations were performed 6 h after LPS-instillation. All mice were sacrificed 24 h post LPS exposure. RA + GA denotes the co-administration of RA and GA as a mixture.

### 4.10. BALF Collection and Inspiratory Capacity Determination

Twenty-four hours post LPS treatment, mice were intratracheally intubated and the peak inspiratory pressure (P_IP_) was measured through the VentElite Small Animal Ventilator (3 cmH_2_O Positive end-expiratory pressure (PEEP), 200 μL tidal volume) (Harvard Apparatus, Holliston, MA, USA). For BALF collection, PBS (1 mL) was injected intratracheally, and the fluid was re-aspirated (lavaged) three times. Cell content in the BALF was measured on a Neubauer hemocytometer as previously described [20]. Protein concentration was measured via the Lowry assay with bovine serum albumin (BSA) used as a standard as previously described [103].

### 4.11. Histopathologic Evaluation

Lung tissue samples were fixed in a 4% paraformaldehyde (PFA) solution for 24 h. Tissues were then dehydrated with ethanol and their morphological changes were analyzed in sliced 5 μm sections under light microscope (Nikon Eclipse 80i, Nikon Corp, Tokyo, Japan) following Hematoxylin and Eosin (H&E) staining as previously described [104]. Lung histologic alterations were examined by an experienced blinded pathologist based on four main parameters of ARDS pathogenesis: hemorrhage, inflammatory cell infiltration, broncheoli injury and vascular intima hypertrophy [29,105]. A score from 0 to 3 (absent, mild, moderate, severe) was assigned to each one.

### 4.12. Determination of MDA Levels in BALF and Lung Tissues

Oxidative stress was determined based on the concentration of the lipid peroxidation marker malondialdehyde (MDA) [106]. Blood samples were collected 24 h post LPS treatment, centrifuged (5000× rpm for 15 min at 25 °C) and stored at −80 °C. MDA levels were measured in BALF and lung tissues via the Oxford Biomedical Research Colorimetric Assay for lipid peroxidation as previously described [107]. For lung MDA content, lung tissues that were snap-frozen in liquid nitrogen and stored at −80 °C until the assay were next pulverized and extracted in an ice-cold 20 mM Tris-HCl buffer (pH = 7.4). A centrifugation at 3000× *g* for 10 min at 4 °C followed, and 400 μL of the collected supernatant was mixed with a 1300 μL N-methyl-2-phenyl-indole (10.3 mM in acetonitrile, ACN) and a 300 μL 12N HCl. Then, the samples were incubated at 45 °C for 1 h and centrifuged at 3500× rpm for 15 min at 4 °C. For the estimation of the protein concentration of the supernatants, the Lowry method was employed (DC protein assay, BIORAD, Watford, UK), and the MDA content was determined spectrophotometrically at 586 nm and expressed in μmol/mg after normalization to total protein levels.

### 4.13. Western Blot Analysis

The potential mechanisms and pathways involved in LPS-induced ARDS were investigated via Western blotting as described previously [103]. Briefly, lung tissues were grinded manually into homogenized fine powder using dry ice, and a lysis buffer was added (1% Triton X-100, 20 mM Tris pH 7.4, 150 mM NaCl, 50 mM NaF, 1 mM EDTA, 1 mM EGTA, 1 mM glycerol phosphatase, 1% sodium dodecyl sulfate (SDS), 100 mM phenylmethanesulfonyl fluoride, and 0.1% protease phosphatase inhibitor cocktail). Protein concentration of each lysate was determined via the Lowry method with BSA used as a standard. Samples were then prepared with Dave’s buffer (4% SDS, 10% 2-mercaptoethanol, 20% glycerol, 0.004% bromophenol blue, 0.125M Tris/HCl), lysis buffer and equal amount of protein, boiled for 10 min at 100 °C, and separated by 10–15% sodium dodecyl sulfate—polyacrylamide gel electrophoresis (SDS-PAGE). They were then transferred onto polyvinylidene difluoride (PVDF) membranes and blocked with 5% nonfat dry milk in a Tris-buffered saline (TBS)/Tween 20 buffer.

Membranes were incubated overnight at 4 °C with the following primary antibodies: VCAM-1 (Cell Signaling Technology, Beverly, MA, USA, Rabbit mAb #39036), ICAM-1 (Abcam, Cambridge, UK, Mouse mAb ab171123), iNOS (Cell Signaling Technology, Rabbit mAb #13120), MyD88 (Cell Signaling Technology, Rabbit mAb #4283), POP (Abcam, Rabbit pAb ab246978), pAkt (S473) (Cell Signaling Technology, Rabbit mAb, #9271), tAkt (Cell Signaling Technology, Rabbit mAb, #9272), Bcl-xL (Cell Signaling Technology, Rabbit mAb #2764), Bax (Cell Signaling Technology, Rabbit mAb #2772), Cleaved caspase-3 (Asp175) (Cell Signaling Technology, Rabbit mAb #9664) and GAPDH (Cell Signaling Technology, Rabbit mAb, #2118). Then, a secondary goat anti-mouse (Cell Signaling Technology, #7076) and a goat anti-rabbit HRP (Cell Signaling Technology, #7074) (Cell Signaling Technology, Beverly, MA, USA) were added, and membranes were incubated for 1–2 h at room temperature. Blots were developed using GE Healthcare ECL Western Blotting Detection Reagents, and densitometric analysis was performed with ImageQuant LAS 500 (Thermo Fischer Scientific Inc., Waltham, MA, USA). Relative densitometry was determined using a computerized software package (Image J 1.53t, NIH, USA), and expression was illustrated as the graph of the relative integrated optical density (IOD) of each treatment group. The values of phosphorylated proteins (phospho-proteins) were normalized to the values of total respective proteins. GAPDH was used as a loading control.

### 4.14. Statistical Analysis

All experimental values are presented as the mean ± standard error of the mean (S.E.M.). Individual treatment groups were analyzed by student’s t-test and comparison among groups was performed using one-way ANOVA followed by a Tukey’s post hoc test for statistical significance using statistic software GraphPad Prism 8 (GraphPad Software, Inc., San Diego, CA, USA). A probability (*p)* value of ≤ 0.05 was considered statistically significant (* *p* < 0.05, ** *p* < 0.01, *** *p* < 0.001). All experiments were performed at least in triplicate.

## Figures and Tables

**Figure 1 ijms-24-14235-f001:**
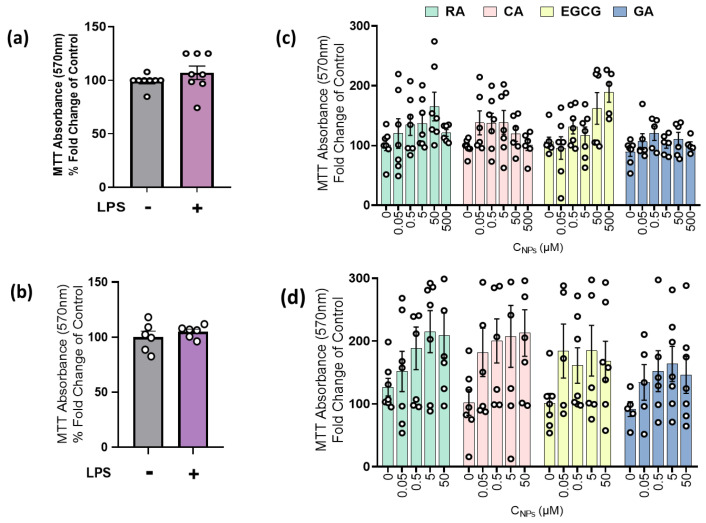
Natural products (NPs) and lipopolysaccharide (LPS) did not affect cell viability of RAW264.7 macrophages and primary mouse lung epithelial cells (MLECs). Evaluation of cell viability following NPs and/or LPS administration in (**a**) RAW 264.7 (n = 8) and (**b**) MLECs cells (n = 6) via the MTT assay. Cells were either untreated (−LPS) or stimulated with LPS (10 μg/mL) for 24 h (+LPS). MTT absorbance was measured at 540 nm. (**c**) Effect of NPs on cell viability of RAW 264.7 macrophages (n = 7). Cells were treated with different concentrations (0–500 μM) of either rosmarinic acid (RA, green), chicoric acid (CA, pink), epigallocatechin-3-gallate (EGCG, yellow) or gallic acid (GA, blue) for 24 h. (**d**) Effect of the co-administration of NPs and LPS (10 μg/mL) on cell viability of RAW 264.7 macrophages (n = 7). Cells were treated with different concentrations (0–50 μM) of either RA + LPS, CA + LPS, EGCG + LPS or GA + LPS for 24 h. For (**a**–**d**), MTT absorbance was measured at 540 nm. Results are expressed as the mean ±S.E.M. and were analyzed via unpaired t-test and one-way ANOVA.

**Figure 2 ijms-24-14235-f002:**
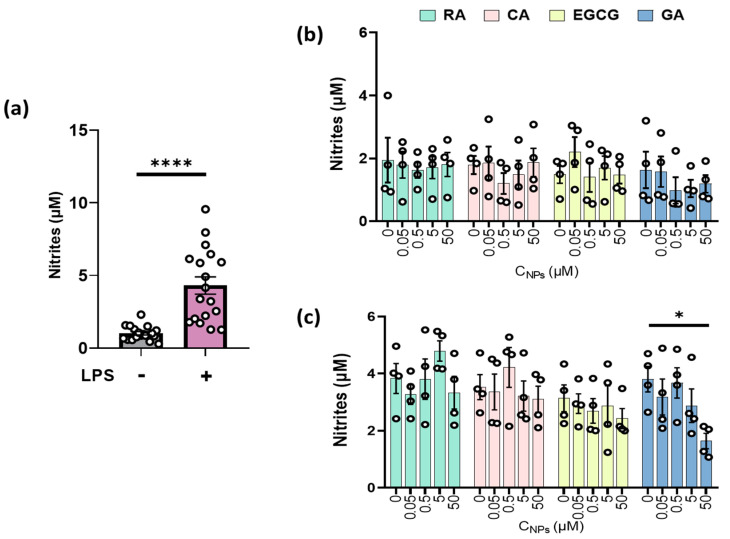
Estimation of the effect of NPs on the LPS-induced nitrite production of unstimulated and stimulated RAW 264.7 cells. (**a**) Evaluation of LPS on nitrite production in RAW 264.7 macrophages. Cells were either untreated (−LPS) or stimulated by LPS (10 μg/mL) for 24 h (+LPS) (n = 18). (**b**) Effect of NPs (0–50 μM) on nitrite concentration (μM) of RAW 264.7 macrophages. Cells were treated with different concentrations (0–50 μM) of either rosmarinic acid (RA, green), chicoric acid (CA, pink), epigallocatechin-3-gallate (EGCG, yellow) or gallic acid (GA, blue) for 24 h (n = 4). (**c**) Effect of NPs (0–50 μM) on nitrite concentration (μM) of LPS-stimulated RAW 264.7 macrophages. Cells were treated with different concentrations (0–50 μM) of either RA + LPS, CA + LPS, EGCG + LPS or GA + LPS for 24 h (n = 4). For (**a**–**c**), nitrite production was measured photometrically in the culture medium. Results are expressed as mean ±S.E.M. of nitrite concentration and were analyzed via an unpaired t-test for (**a**) and one-way ANOVA followed by Tukey’s post hoc test for (**b**,**c**). * denotes *p* < 0.05 and **** denotes *p* < 0.0001 versus the control group.

**Figure 3 ijms-24-14235-f003:**
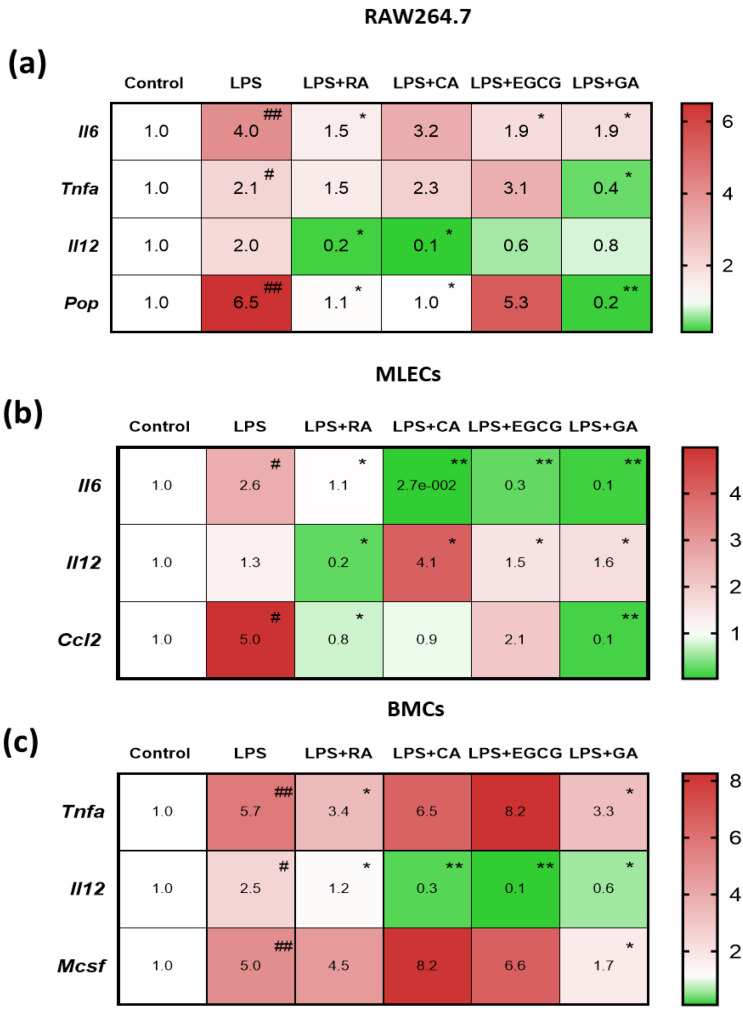
Natural products diminished LPS-elevated levels of key pro-inflammatory mediators. Heat map of gene expression levels of inflammatory markers after Real-Time PCR analysis on (**a**) macrophage RAW 264.7 cells, (**b**) primary mouse lung epithelial cells (MLECs) and (**c**) primary mouse bone marrow cells (BMCs). Cells were incubated with LPS (10 μg/mL) with or without NPs (50 μM) for 24 h. Green color indicates treatment groups demonstrating down-regulation of genes and lower levels of expression than the control groups (<1.0) and red color indicates groups demonstrating up-regulation of genes and higher levels of gene expression than the control groups (>1.0) (n = 6). * denotes *p* < 0.05, ** denotes *p* < 0.01, when compared to the LPS group. # denotes *p* < 0.05 and ## denotes *p* < 0.01 when compared to the control group.

**Figure 4 ijms-24-14235-f004:**
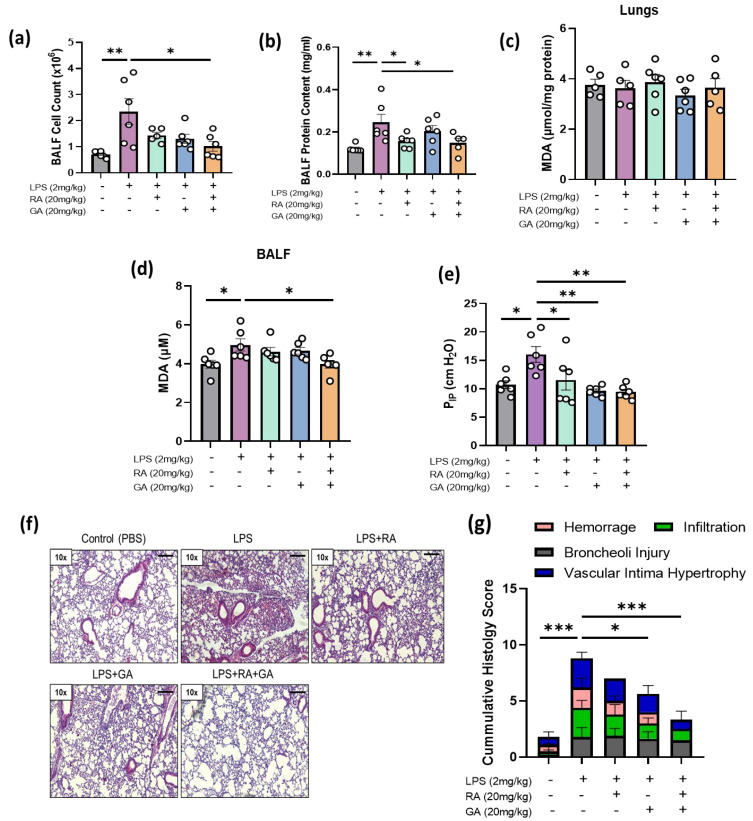
Administration of natural products restored LPS-induced inflammation and oxidative stress in bronchoalveolar lavage fluid (BALF) and ameliorated the LPS-induced lung injury and respiratory capacity of mice. Mice were administered with either LPS alone (2 mg/kg), LPS + RA, LPS + GA (20 mg/kg) or LPS with a mixture of RA + GA (20 mg/kg). Administration of RA, GA or RA + GA was carried out 6 h after LPS stimulation. BALF was collected 24 h after treatment and cell count (**a**) and protein concentration (**b**) were measured. Malondialdehyde (MDA) levels were measured as a marker of oxidative stress in lung tissue (**c**) and BALF (**d**). Graphical representation (**e**) of the peak inspiratory pressure/P_IP_ (cmH_2_O) after administration of RA, GA and their mixture on LPS-challenged mice. Mice were given either LPS alone (2 mg/kg), LPS + RA (20 mg/kg), LPS + GA (20 mg/kg) or LPS with a mixture of RA + GA (20 mg/kg). RA, GA or RA + GA administration was carried out 6 h after LPS stimulation. Each animal was anaesthetized and ventilated with 3 cmH_2_O positive end-expiratory pressure (PEEP) and a tidal volume of 200 μL. Representative images (**f**) of lung tissues as visualized under 10x light microscopy after Hematoxylin and Eosin (H&E) staining and related graph bars (**g**). Histologic changes were quantified as cumulative histology scores and measured based on four markers distinctive of ARDS pathogenesis: hemorrhage, infiltration, broncheoli injury and vascular intimal hypertrophy. Results are expressed as mean +/− S.E.M. (n = 6) and analyzed via one-way ANOVA, followed by Tukey’s post hoc test. For statistically significant results, * denotes *p* < 0.05, ** *p* < 0.01 and *** *p* < 0.001 when compared with the control group.

**Figure 5 ijms-24-14235-f005:**
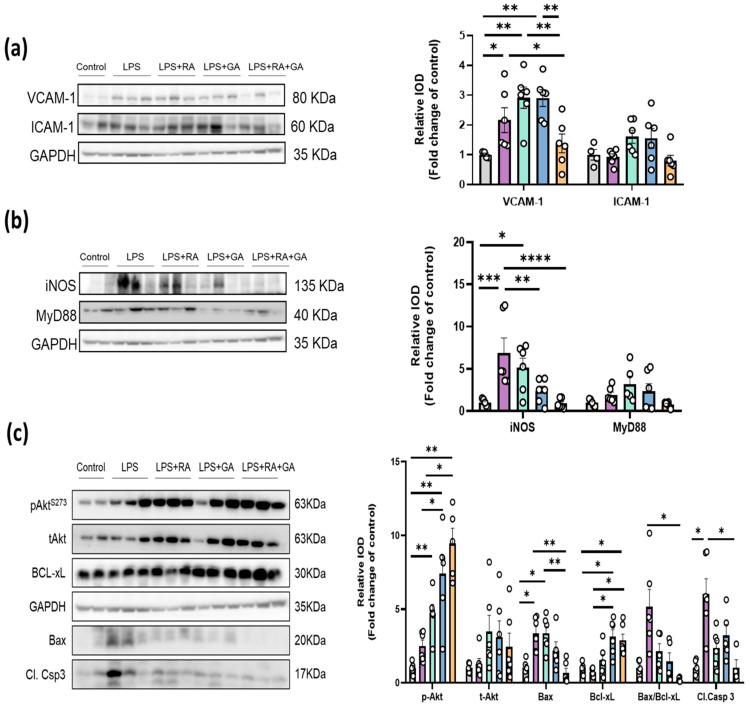
Co-administration of RA and GA abrogated inflammatory signaling and apoptosis in the lungs in LPS-induced ARDS in vivo. Representative Western blots and relative densitometric graphs for VCAM-1 and ICAM-1 (**a**) and iNOS and adaptor protein MyD88 (**b**) are shown after normalization to GAPDH protein, which was used as a loading control. (**c**) Representative Western blots and relative densitometric graphs for protein expression in lung tissues of pAkt (protein kinase B) after normalization to total protein, and tAkt, B-cell lymphoma-extra-large (Bcl-xL), BCL2- Associated X (Bax), Bax/Bcl-xL and cleaved caspase-3 after normalization to glyceraldehyde 3-phosphate dehydrogenase (GAPDH) protein (35 KDa). (**d**) Co-administration of RA and GA restored protein expression levels of the POP enzyme in lung tissues in vivo following treatment with LPS. Representative Western blots and the relative densitometric graph for protein levels of prolyl oligopeptidase (POP) after normalization to GAPDH protein, which was used as loading control. Results are expressed as mean +/− S.E.M. (n = 6) and analyzed via one-way ANOVA, followed by Tukey’s post hoc test for statistically significant results. * denotes *p* < 0.05, ** *p* < 0.01, *** *p* < 0.001 and **** *p* < 0.0001 when compared with the control group. Dots represent biological replicates.

**Table 1 ijms-24-14235-t001:** Source of origin and chemical characteristics and formulas of the four natural products used in our research (RA, CA, EGCG, GA) [14,20,93].

Natural Product	Chemical Formula
Rosmarinic acid (RA) Ester of caffeic acid	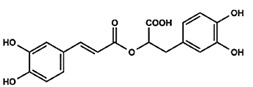
Chicoric acid (CA) Hydroxycinnamic acid, phenylpropanoid	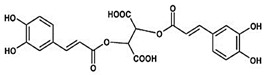
Epigallocatechin-3-gallate (EGCG) Ester of epigallocatechin and gallic acid	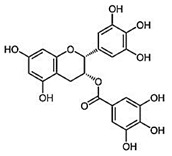
Gallic acid (GA) Trihydroxybenzoic acid	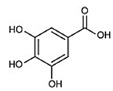

**Table 2 ijms-24-14235-t002:** Sequences and product size of the primers used for Real-Time PCR analysis.

Primer	Forward Sequence (5′–3′)	Reverse Sequence (5′–3′)	Product Length
β-actin	GCAAGCAGGAGTACGATGAGT	AGGGTGTAAAACGCAGCTCAG	88
Il-6	AGTCCTTCCTACCCCAATTTCC	TGGTCTTGGTCCTTAGCCAC	80
Il-12α	CAAGGATATCTCTATGGTCAGCGT	GGTAGCGTGATTGACACATGC	95
Tnf-α	ATGGCCTCCCTCTCATCAGT	TGGTTTGCTACGACGTGGG	100
M-csf	CCTTCTTCGACATGGCTGGG	GTTCTGACACCTCCTTGGCA	82
Pop (Prep)	ACCTCCGTGCAGGAGTATCA	GTGCCTCCACGAAAGCCTTA	97
Ccl2	CCAATGAGTAGGCTGGAGAGC	GAGCTTGGTGACAAAAACTACAGC	81

## Data Availability

Data available on request due to privacy restrictions.

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
