# Peer review of "Synergistic Pulmonoprotective Effect of Natural Prolyl Oligopeptidase Inhibitors in In Vitro and In Vivo Models of Acute Respiratory Distress Syndrome"

_ijms, 2023, doi:10.3390/ijms241814235_

Round 1

Reviewer 1 Report

The study is well-designed and properly conducted and the results are clearly presented. I have few comments:

- decreased production of NO is presented as a protective effect of GA, but the actions of NO on the lung are numerous and NO administration has been shown to ameliorate neutrophilic inflammation in other models. Could the Author expand on this?

- Figure 1b: the numbers on the x axis are not entirely visible

- the discussion should contain a paragraph on the study limitation and potential problems with translation of the results

Author Response

Reviewer #1:

The study is well-designed and properly conducted and the results are clearly presented. I have few comments:

We would like to thank the Reviewer for evaluating our manuscript. During this revision process, we have addressed all additional comments point by point, which helped us to improve our manuscript.

Comment 1

Decreased production of NO is presented as a protective effect of GA, but the actions of NO on the lung are numerous and NO administration has been shown to ameliorate neutrophilic inflammation in other models. Could the Author expand on this?

We thank the reviewer for the important comments aiming at the improvement of our manuscript. The pulmonoprotective effect of NO is undebatable, however the excessive NO production via endothelial NO synthase (eNOS) or inducible NO synthase (iNOS) is also reported to lead to reactive nitrogen species (RNS) formation, nitrooxidative stress and finally to deleterious effects in the lungs (PMID: 18236016). In our study, we employed NO production measurement assay to estimate indirectly the macrophage polarization to M1 pro-inflammatory phenotype (PMID: 19478557) after LPS administration and the anti-inflammatory properties of our natural products. GA exerted the optimal effects.  

We added the following text in the Discussion section in order to expand on the role of NO in the lungs as requested by the reviewer. Please find the respective information in Page 11, lines 308-326 in the revised manuscript as follows:

“Although, nitric oxide (NO) is an important part of the host defense mechanism and plays physiological roles in several cellular processes, the formation of RNS after excessive NO production, is considered the main NO contribution in both acute and chronic inflammation [35]. NO might have distinct roles in different stages of pulmonary inflammatory diseases, as it has been reported to be deleterious and pro-inflammatory in acute and severe stages of the disease but also acting as a vasodilator with potentially protective and anti-inflammatory properties in more stable conditions. These contradictory effects of NO may be due to the localization, time and the amount of NO synthesis, the acute or chronic character of the immune response or the type of lung injury model and the absence of specificity for the different NOS isoforms inhibitors used within the studies [36]. Moreover, despite the alleged harmful effects of NO that have been demonstrated, NO inhalation has been suggested as a potential therapeutic treatment for ARDS patients in order to reduce neutrophilic inflammation and improve overall gas exchange by acting directly to the pulmonary vasculature located in the vicinity of the ventilated lung. However, results of NO efficacy at a clinical level are controversial, as it is associated with limited to no improvement in oxygenation and mortality rates of ARDS patients [37]. Therefore, additional pre- and clinical studies are required in order to decipher the complete pulmonoprotective or deleterious potential of NO in lung diseases.

Comment 2

Figure 1b: the numbers on the x axis are not entirely visible

We thank the reviewer for their comment. We changed Figure 1b so that the numbers on the x axis are visible. Please find the revised Figure 1 in the revised manuscript in page 3.

Comment 3

The discussion should contain a paragraph on the study limitation and potential problems with translation of the results

We thank the reviewer for their comment. We added a new paragraph in the Discussion section. Please refer to page 15, lines 509-523 in the revised manuscript as follows:

“A limitation of our study is that we could not correlate the in vivo doses and the in vitro tested concentrations of each natural product, whereas dose and concentration regimens were based on previous reports [20,89]. Moreover, our study lacks clinical proof of the pulmunoprotective effect of the compounds reported, or their combination. The better assessment of the pharmacokinetics and pharmacodynamics of our compounds could improve our knowledge concerning the degree of contribution of the two NPs in vivo and probably also shed some light on their synergistic activity. Moreover, since the existing murine lung injury models do not seem to precisely recapitulate all aspects of the human ARDS, the challenge remains to translate these findings and prove the efficacy of our compounds in clinical trials. Even though translatability of animal studies has been unsuccessful in ARDS-related clinical trials until now [73,90], the better identification of the biological pathways involved, the development of novel approaches in NPs administration for improved bioavailability and the selection of an appropriate patient population based on evidence-driven criteria, such as biomarkers, risk factors, phenotypes or concurrent medications of the patients, could be helpful in the achievement of this objective”.

Reviewer 2 Report

The presented study evaluated the pulmonoprotective effects of natural prolyl oligopeptidase (POP) inhibitors in vitro and in vivo. Four natural compounds were selected due to their anti-inflammatory and anti-oxidant effects. RAW264.7, primary murine lung epithelial and bone barrow cells were co-stimulated with LPS and the four natural products. The most profound reduction of pro-inflammatory mediators was observed after rosmarinic and gallic acid co-treatment. LPS-challenged mice received a a single therapeutic treatment with these natural products or a combined treatment. The combined treatment demonstrated the most beneficial effects on ARDS pathophysiologic mediators and histology. 

This is an interesting an promising study for the supportive treatment of pulmonary inflammation. Methodology and selected pathophysiologically relevant mediators are well chosen and presented.

p1, l. 27 and l33: abbreviations NP and MDA not introduced in abstract

p2/24, l. 47-48:  Since the Berlin definition of ARDS (doi: 10.1001/jama.2012.5669) came into effect this differentiation between ARDS/ALI is outdated.

p5, l. 157 and 6/24, l. 177: the NP concentration used in the presented gene expression results is not mentioned. 

Minor editing recommended for the discussion:

p. 11, l.303

p.11, l. 305

Author Response

Reviewer #2:

The presented study evaluated the pulmonoprotective effects of natural prolyl oligopeptidase (POP) inhibitors in vitro and in vivo. Four natural compounds were selected due to their anti-inflammatory and anti-oxidant effects. RAW264.7, primary murine lung epithelial and bone barrow cells were co-stimulated with LPS and the four natural products. The most profound reduction of pro-inflammatory mediators was observed after rosmarinic and gallic acid co-treatment. LPS-challenged mice received a single therapeutic treatment with these natural products or a combined treatment. The combined treatment demonstrated the most beneficial effects on ARDS pathophysiologic mediators and histology. 

This is an interesting and promising study for the supportive treatment of pulmonary inflammation. Methodology and selected pathophysiologically relevant mediators are well chosen and presented.

We thank the reviewer for their evaluation, their important comments and their time and consideration facilitating the improvement of our manuscript. Please find below the response to the reviewer’s comments

Comment 1

p1, l. 27 and l33: abbreviations NP and MDA not introduced in abstract

We added abbreviations for NP and MDA in abstract as requested by the reviewer. Please refer to the revised manuscript in page 1 lines 27 and 34 respectively.

Comment 2

p2/24, l. 47-48:  Since the Berlin definition of ARDS (doi: 10.1001/jama.2012.5669) came into effect this differentiation between ARDS/ALI is outdated.

We thank the reviewer for their important comment. Following reviewer’s instruction, we changed the text. Please refer to the revised manuscript in page 2, lines 47-50 as follows:

“Over the years and according to the Berlin definition of ARDS, ARDS has been recognized as significant pulmonary complication with a global impact [2] and associated with considerable rates of morbidity and mortality [3,4] without effective pharmacological therapies [1,2].

Comment 3

p5, l. 157 and 6/24, l. 177: the NP concentration used in the presented gene expression results is not mentioned.

We thank the reviewer for the important comment. We added the concentration of NPs in the Results section and in the legend of Figure 3. Please refer to the revised manuscript in page 5, line 160 and page 6, lines 182-183.

Comment 4

Minor editing recommended for the discussion: p. 11, l.303 - p.11, l. 305

We replaced previous text with this sentence in Discussion section. Please refer to the revised manuscript in page 11, lines 331-333:

“The mature neutrophils, that remain as a reserve in the bone marrow and released upon the inflammatory stimulus, are the first cells recruited from circulation to the site of infection [40].”

Reviewer 3 Report

1) Introduction is very clear. No comments.

2) I am usual to insert materials and methods soon after the introduction. Move this section, is not required as per Journal Style.

3) Materials and methods: I suggest to insert the Ethical committee approval (if required by your regulation), and that the study was conducted according to the required standards for animals.

4) Results and discussion are clear.

5) please add in the discussion possible future impact of your research on the clinical management of ARDS patients. This would increase clarity and interest for clinicians.

Author Response

Reviewer #3:

We thank the reviewer for their fruitful comments helping us improve our manuscript. Please find below a point-by point response to the reviewer’s concerns.

Comment 1

I am usual to insert materials and methods soon after the introduction. Move this section, is not required as per Journal Style.

We thank the reviewer for their comments. According to the template of MDPI journals the sequence of the sections should be 1. Introduction, 2. Results, 3. Discussion, 4. Materials and methods. Therefore, we cannot, unfortunately move this section.

Comment 2

Materials and methods: I suggest to insert the Ethical committee approval (if required by your regulation), and that the study was conducted according to the required standards for animals"

We thank the reviewer for noticing this omission from our manuscript. An ethical approval for conducting the in vivo experiments in the study is now included in the Materials and Methods section. Please refer to the revised manuscript in page 18, lines 640-646:

“A total of 30 (13–14 weeks of age) C57BL/6J male mice (~25 g) were bred and housed at 25°C in microisolators in the Animal Facility of Biomedical Research Foun-dation Academy of Athens and were given food and water ad libitum. Experiments were carried out in accordance with the “Guide for the care and use of Laboratory animals” and with the Directive 2010/63/EU and experiments were approved by the Ethics Committee, Veterinary Service of the Prefecture of Athens (Approval No: 124357/15-02-2022) according to ARRIVE guidelines [99].”

Comment 3

Please add in the discussion possible future impact of your research on the clinical management of ARDS patients. This would increase clarity and interest for clinicians.

We thank the reviewer for the comment. We added the following text in the Discussion section in page 15, lines 529-540:

“Taking under consideration that clinical arsenal lacks specific prophylactic and therapeutic therapies against ALI/ARDS, we herein propose a novel combination of NPs, that of RA and GA as a potent therapeutic intervention after the establishment of the disease. The use of NPs in therapy, based on their cytotoxicity profile, might also contribute to the mitigation of side effects, such as severe gastrointestinal irritations, allergic reactions or immunosuppression (such as steroids and immunotherapy [48,91,92]), while it could be beneficial against long-term morbidity after critical illness. In this study, for the first time we have proposed a new dose and administration regimen for RA and GA, this of intraperitoneal injection, that can be employed in future clinical studies investigating their pulmunoprotective potential. Nevertheless, further pre- and clinical studies are required for the establishment of the combination therapy as a potent therapeutic intervention against ALI/ARDS”.